

# Systematic analysis of virga and its impact on surface particulate matter observations

Nakul N. Karle[1], Ricardo K. Sakai[1], Rosa M. Fitzgerald[2], Charles Ichoku[1], Fernando Mercado[3], William R. Stockwell[2]

[1]Program in Atmospheric Sciences, Howard University, Beltsville Campus, MD, 20705, USA
[2]Department of Physics, The University of Texas at El Paso, El Paso, TX, 79968, USA
[3]Texas Commission on Environmental Quality, Austin, TX, 78711, USA

*Correspondence to*: Nakul N. Karle (nakul.karle@howard.edu)

**Abstract.** Studies focusing on virga are rare, even though it is a commonly occurring phenomenon. In this study, we
investigated aerosol backscatter profiles from a ceilometer located on the University of Texas at El Paso (UTEP) campus from
2015-2021 to identify virga events. Ceilometer data effectively captured virga events from regular precipitation based on the
backscattering intensities. To characterize the virga phenomena, a systematic method was developed using ceilometer profiles,
soundings, surface rain gauges, and radar data from the nearest National Weather Service (NWS). A total of 50 virga events
were identified during the study period. These events appeared only during a specific time of the year, revealing a seasonal
occurrence pattern. We identified and classified these virga events and investigated their impact on the surface measurements
recorded by the on-campus Continuous Ambient Air Monitoring Station (CAMS). Virga events were classified as columnar
and non-columnar events based on their aerosol profiles. We observed that during some of the columnar virga events, surface
PM levels displayed a sudden upward trend indicating aerosol loading in the surface layer after precipitation evaporation.
Twenty of the virga events showed a columnar structure out of the fifty identified in this study. More detailed analysis of
selected events shows that virga affects regional air quality. A significant result of this study is that analysis of sudden changes
in local air quality need to consider the possible effects of virga on the surface layer.

## 1 Introduction

Extreme weather events and climate variability directly impact the hydrological cycle, which affects all life on Earth.
Understanding hydrologic processes require a quantitative description of temporal variability in precipitation. Meteorologists
define precipitation as liquid or solid water that falls from the sky to the ground. However, a type of precipitation known as
virga never reaches the ground. Petterssen in his work, defined virga as the rain that falls from the clouds but evaporates before
making it to the ground (Petterssen, 1958). Virga is also described as a sudden change in the brightness of a precipitation
(water or ice particles) shaft beneath a cloud (Fraser and Bohren, 1992). It is a commonly observed phenomenon in hot, arid
regions. According to Ludlam, the cumulus cloud bases in desert regions are higher above the ground (3-4 km), and that causes
the precipitation on dry days to evaporate completely before even reaching the ground (Ludlam, 1980).



Water has a high heat of vaporization (40.65 kJ/mol (540 calories $g^{-1}$)), so the transition from liquid to gas requires a significant amount of heat energy (Henderson-Sellers, 1984). Precipitation evaporating at a high altitude can heat up as it falls, due to adiabatic compression, resulting in a gusty downburst that can significantly and rapidly warm the surface temperature. The abrupt changes in the lower troposphere therefore can have a significant impact on weather forecasting, climate prediction, aviation, local air quality, and vegetation. Singh and O'Neil reviewed earth's climate system using the second law of thermodynamics (Singh and O'Neill, 2022). However, the thermodynamics involved during the virga events is not entirely known. During virga, the rain changes from liquid to vapor form, removing significant amounts of heat from the air and causing various weather effects. Colder air parcels can descend quickly to the ground, causing wet or dry microbursts that can pose extreme danger to small planes and other aircraft. A well-known example is the tragic crash of Eastern Airlines Flight 66 on June 24, 1975, while approaching New York's John F. Kennedy International airport. Fujita (Fujita and Byers, 1977) coined the term "downburst" to describe the induced wind shear that affected the airport. The Joint Airport Weather Studies (JAWS) project, designed to study the three-dimensional structure of microbursts in space and time, discovered that some microburst events were associated with virga shafts with little or no rain on the surface (McCarthy et al., 1982). Wilson et al. claimed that dry microbursts and wet microbursts exist, each with a different forcing mechanism, and they associated dry microbursts with virga events using doppler radar measurements from the JAWS project (Wilson et al., 1984).

Even though virga is a captivating visual phenomenon with broader research implications, scientific publications on the subject are scarce. Previous studies investigated it using remote sensing instruments such as radar and lidar, both ground-based, airborne, and  satellite observations. (Wang et al., 2018) quantified global virga using spaceborne radars. They showed that it accounted for about 50% and 30% of overall false precipitation events detected by the Tropical Rainfall Measuring Mission (TRMM) Microwave Imager and Global Precipitation Measurement Microwave Imager, respectively, in arid regions. Using long-term measurements from the TRMM's Precipitation Radar data over India and the surrounding oceans, (Saikranthi et al., 2014) discovered significant virga occurrences (20% and 14%) in the dry, semi-arid regions of Northwest India and Southeast Peninsular India, respectively. They also discovered the highest occurrence of virga in India during the pre-monsoon. (Airey et al., 2021) derived the characteristics of desert precipitation in the UAE using a dataset of ceilometer observations spanning two years. They discovered that 28 of the 105 rain-producing events were virga, with small droplets, high cloud bases, reduced cloud depths, and cold cloud bases as multiple regional contributing factors. The authors also highlighted the significance of understanding the amount of precipitation in drought-prone arid regions with limited water resources for agriculture, irrigation, and domestic usage.

Cheng and Yi used ground-based lidars to observe mixed-phase virga from a thin supercooled liquid layer cloud base on 20 occasions. They discovered that the ice crystal particles in these virga cases had smaller mean diameters and narrower size distributions as altitude increased (Cheng and Yi, 2020). Beynon and Hocke detected and studied snow virga in Bern, Switzerland using a Doppler Micro Rain Radar (MRR). In their work, they specifically focused on a 22-hour long snow virga event from 17 March 2013 (Beynon and Hocke, 2022). The authors concluded that the 22-hour virga was caused by prevailing wind shear, which carried moisture-saturated air in the upper air layers over the measuring station while the wind blew in the




lower air layers, carrying unsaturated air with it. The authors also discovered a discrepancy between the MRR observations and the ERA-5 dataset, a global atmospheric reanalysis produced at the European Centre for Medium-Range Weather Forecasts by the Copernicus Climate Change Service (C3S) (ECMWF). Unlike MRR, which did not record any ground precipitation during the snow virga event, the latter showed a drizzle on the ground for 4 hours. The preceding work also highlights the importance of high resolution and frequency radiosonde, lidar, radar, and radiometer observations for model validation or data

assimilation (Beynon and Hocke, 2022). Similar snow virga has been detected and observed by (Jullien et al., 2020) and (Grazioli et al., 2017) in the Antarctic region. Virga was even linked with severe climatic events such as droughts. Evans et al. used radar data to classify precipitation over three locations in the Canadian prairie during the 1994-2004 droughts. They classified the precipitation over the drought region as convective, stratiform, or virga. Virga with an average cloud base temperature greater than $0^0$C resulted in efficient sublimation loss of precipitation that contributed to decreased surface

precipitation (Evans et al., 2011).

   While all the above studies clearly advanced knowledge and understanding of regional virga detection and its variations to a great extent, some critical questions remain unanswered. Although heterogeneous-components and dissolved aqueous-phase constituents are released when cloud water or falling rain droplets (hydrometeors) either sublimate or evaporate, the contribution of this material towards local air quality is not yet well quantified. According to (Tost et al., 2006), during this

evaporation or sublimation process, any non-ionic, volatile compounds are converted to gas, scavenged by aerosol particles and redistributed into aerosols. Non-volatile components such as chlorides, sulfates and similar ionic compounds could serve as condensation nuclei for new aerosol particles. The quantification of the relationship between virga and the aerosol loading in the lower troposphere is an important new research question for atmospheric chemistry.

   El Paso with a semi-arid climate is a border city at the western tip of Texas, USA. It borders with the Mexican city Ciudad

Juarez and together this region hosts two large airports and several small aerodromes. As a result, it is critical to research regional virga events thoroughly. The current study focuses on two aspects of virga events observed in this region. First is the characterization of virga seasonal pattern for the region and second is the analysis of ground measurements during the virga event. This region hosts a ceilometer located at the University of Texas at El Paso campus and has been functional since 2015. Virga events were detected primarily using the aerosol backscatter profiles from the ceilometer. El Paso is known for its high

ozone events during summer (Karle et al., 2020) and the high PM events during winter (Lara et al., 2022; Fitzgerald et al., 2021). Because virga's impact on local air quality has yet to be published, we present our observations, analysis, and discussion of case studies of some of the events observed in El Paso.

   The following is how this paper is structured. In Section 1, we present an introduction to previous studies and their significance. Section 2 covers the instrumentation and datasets used and the data collection method; Section 3 discusses the criterion for

selecting virga events for this study. Section 4 examines the seasonal pattern of virga events in this region, followed by two case studies. Section 5 discusses and concludes our findings.



## 2 Dataset and Methodology

Various datasets were used in this study to conduct a thorough analysis and consolidate observations and scientific claims to build substantial research. Dataset used in this work includes the ceilometer observations, vertical profiles from radiosonde, and doppler weather radar, and ground-based air quality in-situ measurements.

### 2.1 Site description

The city of El Paso (latitude: $31^0 47' 20"$; longitude: $-106^0 25' 20"$; elevation: 1145 m a.s.l.) is in the far western corner of Texas, separated by the Rio Grande River from the Mexican city of Juárez and surrounded by the Chihuahuan desert. El Paso has a semi-arid climate characteristic of the urban southwestern US climate. It is also known as "Sun City" because of its approximately 302 days of sunshine annually. The Sun city's monsoon season starts on 15 June and runs through the end of September. The rainy period of the year lasts for around 4 months, from mid-June to the end of October, with a sliding 31-day rainfall of at least 0.5 inches. The month with the most rain in El Paso is August, with an average rainfall of 1.5 inches. The rainless period of the year lasts for around 7-8 months, from the end of October to first week of June. The month with the least rain in El Paso is April, with an average rainfall of 0.2 inches (Karle, 2021).

### 2.2 Instrument

#### 2.2.1 Ceilometer CL31 and CL51

Based on attenuated aerosol backscatter profile measurement, the ceilometer is an essential instrument in both robust functionality and cost efficiency in detecting cloud cover, cloud base height and aerosol layer height which can be used to infer the planetary boundary layer (PBL). A ceilometer located at the University of Texas at El Paso (UTEP) campus is used to measure the intensity of backscatter caused by precipitation, clouds, fog, and the haze and creates profiles of signal strength vs. height measured over time. Two Vaisala ceilometer models, CL31 and CL51, were used to monitor the aerosol layer in this region using aerosol backscatter profiles. Both ceilometers are eye-safe single-lens mini-lidar systems that detect cloud base heights and vertical visibility by continuously monitoring aerosol backscatter profiles at $910 \pm 10$ nm (infrared light). CL31 has an InGaAs MOCVD diode with a pulse frequency of 10 kHz and a measurement range of 0-7.7 km. The typical uncertainty of the attenuation of the backscatter coefficient for a 30-minute average duration is 20%.

CL31 was installed on the University of Texas at El Paso (UTEP) campus in March of 2015. It was subsequently replaced, and a new CL51 was commissioned in August 2020. CL51 has a spatial resolution of 10 m and can measure clouds and aerosol layers up to 15,400 m vertically. CL51's diode-laser technology, like the CL31's, is a semiconductor InGaAs diode laser with a wavelength of $910 \pm 10$ nm. The CL51, on the other hand, has a more powerful laser with a pulse energy of 3 µJ, which is higher than the CL31's 1.2 µJ, and a pulse frequency of 6.5 kHz. Furthermore, the instrument's single-lens optics allow it to detect anomalies in a measurement range of approximately 50 m above the ground surface.



### 2.2.2 Data processing

The ceilometer's laser pulses are scattered back by all types of hydrometeors. Although the ceilometer is primarily meant to offer a constant indication of cloud base height, it also keeps a complete record of the returned signal strength as a function of
height. Analysis of the backscatter signal can be in terms of the vertical profiles of the backscattering coefficient and the extinction coefficient using the LIDAR equation. The ceilometer data includes profiles through rain, snow, haze, and strong returns from the cloud base because it operates continually. Rogers et al., in their work, have discussed in detail the extinction and backscattering by these atmospheric constituents (Rogers et al., 1997). The Vaisala proprietary software BL-view (version 2.1.1) was used extensively in this work for data analysis. The software uses a combination of gradient methods along with
built in sky condition algorithms containing five modules to detect and measure cloud base heights and the planetary boundary layer heights (Schafer et al., 2004) from the measured attenuated backscatter profiles. Both the spatial and temporal aerosol backscatter intensity profiles for detection of virga events were obtained by processing them using the BL-view.

### 2.2.3 Next-Generation Radar information

The Weather Surveillance Radar (WSR-88D), developed in 1988 with a doppler capability, is part of the next-generation radar (NEXRAD) network. There are over 158 such WSR-88Ds that operate around the United States. With a power output of 750,000 watts, it is one of the world's most powerful radars. Its 28-foot-diamter antenna inside the dome can gaze at 14 different elevations every 5 minutes. A detailed description of the MSR-88D instrument is available on NWS's webpage ([www.weather.gov/iwx/wsr_88d](www.weather.gov/iwx/wsr_88d)). During the virga occurrences, data from the doppler radar at the NWS in Santa Teresa, New
Mexico, was vital in detecting and confirming precipitation in the air. We obtained and used 5-minute resolution data for each day of the virga event from ([www.ncdc.noaa.gov/nexradinv/](www.ncdc.noaa.gov/nexradinv/)). We extensively used open-source NOAA's National Centre for Environmental Information (NCEI) Radar software ([www.ncdc.noaa.gov/wct/](www.ncdc.noaa.gov/wct/)) for data visualization and analysis.

### 2.3 Vertical profiles from the National Weather Service

Radiosonde observations were obtained from the nearest National Weather Station (NWS) located at Santa Teresa, New
Mexico, a few miles away from the study site. The local NWS launches two radiosondes daily at 0 UTC (UTC −6 hrs MDT and UTC −7 hrs MST) and 12 UTC. Skew-T plots helped us analyse the upper air conditions, especially the temperature and moisture content on the event days. These Skew-T graphs were useful for analysing the thermodynamics of the atmospheric profiles. All the soundings data used in this study was obtained from the University of Wyoming Atmospheric Science Radiosonde Archive database (weather.uwyo.edu/upperair/sounding.html).

### 2.4 Continuous Ambient Monitoring Stations (CAMS)

There are 12 Continuous Ambient Air Monitoring Stations (CAMS) that have been installed in various parts of El Paso for decades. These sites continuously monitor the ambient air and report and issue warnings when pollution anomalies occur.



CAMS 12, located on the UTEP campus (EPA side number 48-141-0037, $31^076'82"$ N, $106^050'12"$ W), is one such station operated and maintained by the Texas Commission on Environmental Quality (TCEQ) El Paso regional office. This station

provided local meteorological data such as temperature, relative humidity, dew point temperature, wind speed, and maximum wind gusts. Usually, CAMS data are provided on an hourly basis and the same was used in this research. The hourly data from the CAMS was used to calculate the rate of change of meteorological variables during virga events to investigate the changes associated with the virga on the surface layer. Later, linear regression was used to determine any correlations between these parameters and investigate virga's impact on them.

**3 Virga events selection criteria**

One of the critical factors for precipitation is the acceptable moisture content in the air between the ground and the cloud base. These humid conditions facilitate the precipitation falling from the clouds to make it to the ground. However, when the layer of air between the cloud base and the ground is dry, precipitation evaporates before reaching the ground. This evaporating precipitation appears as streaks extending down the cloud. Extensive use of the UTEP ceilometer for analysing planetary

boundary layer heights (PBLHs) and their impact on local air quality is well documented (Karle, 2017; Karle et al., 2020; Karle, 2021; Karle et al., 2021).  Several precipitation events with streaks extending down the cloud were observed during the investigation of the aerosol backscatter profiles for PBLH measurements from 2015 to 2021 for these studies. We investigated the events further using ground measurements and other remote sensing data for the region and identified them to be virga occurrences.

Since the literature on virga and detection of its various forms is scarce, we decided to establish our criteria based on a combination of available datasets from various sources. In our method, backscatter intensity profiles from the ceilometer were the primary source of virga detection. Days in which precipitation below the cloud base was observed to fade as it falls towards the ground were chosen. All the selected events had virga detected for at least an hour. There were also instances where precipitation was observed before or after the virga events. National Weather Service (NWS) radiosonde vertical profiles were

used to estimate the cloud base and analyse the dryness in the air near the surface. Radar data from NWS confirmed the presence of precipitation of rain clouds, and finally, using the data from the CAMS located at UTEP, the absence of any precipitation was confirmed. By combining these data sources, we ruled out the possibility that the virga event described in this study was not part of the precipitation that reached the ground in the form of light rain.

Figure 1 depicts an attenuated aerosol backscatter profile obtained from Vaisala CL31 on January 25, 2017, from 00-15 UTC.

The strongest signal (indicated in red) is from precipitation below the cloud base, and the remaining signal (light blue, green and yellow with increasing intensity) is from aerosol backscatter. A streak of precipitation is observed extending downwards from 7:45 UTC to 13:30 UTC. Base reflectivity imagery from the NEXRAD WSR-88D located at National Weather Service (NWS), Santa Teresa, New Mexico (purple dot) is examined (Figure 2). We observe light precipitation at the research site UTEP shown in a red dot (Figure 2). The Base Reflectivity colour over UTEP site at 8:56 UTC correspond to the intensity of



the radiation received by the radar antenna at NWS (KEPZ). A threshold of 20 dBZ is usually the point at which light rain

begins as seen in this case over the study site UTEP (red dot) in Figure 2. However, no precipitation was recorded at the ground

station at UTEP, indicating that the precipitation had not reached the ground, thus confirming the occurrence of virga. The

NWS (KEPZ) vertical profiles at 00 (5 pm MST) and 12 UTC (5 am MST) revealed a large gap between T (temperature in

red) and $T_d$ (dew point temperature in blue) above the surface, indicating dry air (Figure 3). In the 12 UTC profile, at 680 hPa,

we notice the T and Td close to each other, implying moist air, and at 560 hPa, T and $T_d$ overlap, indicating saturated air.

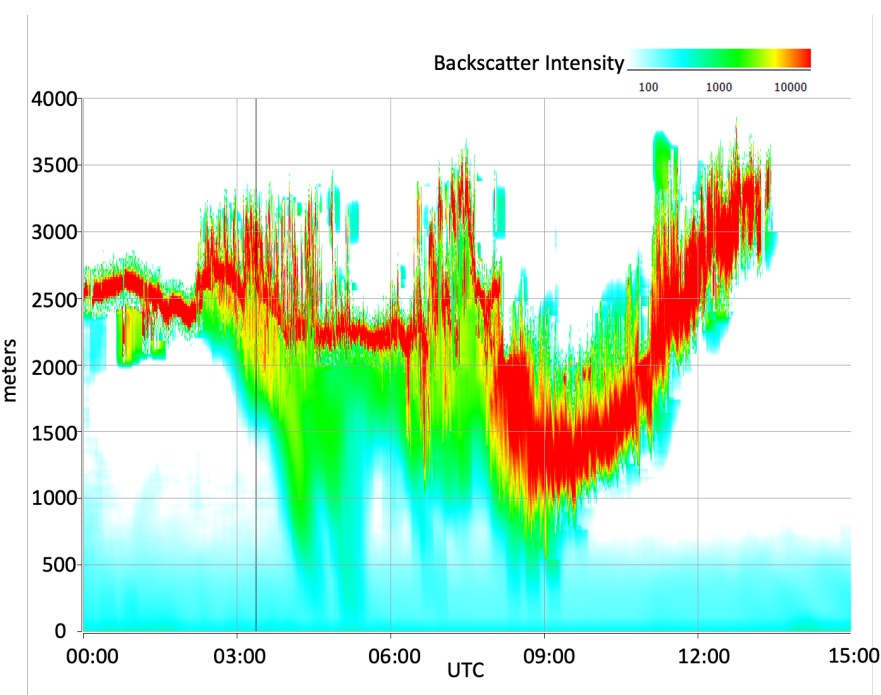

**Figure 1:** Aerosol backscatter profile obtained from CL31 for 25 January 2017 from 00-15 UTC showing virga occurrence from early 8 UTC to 13 UTC.






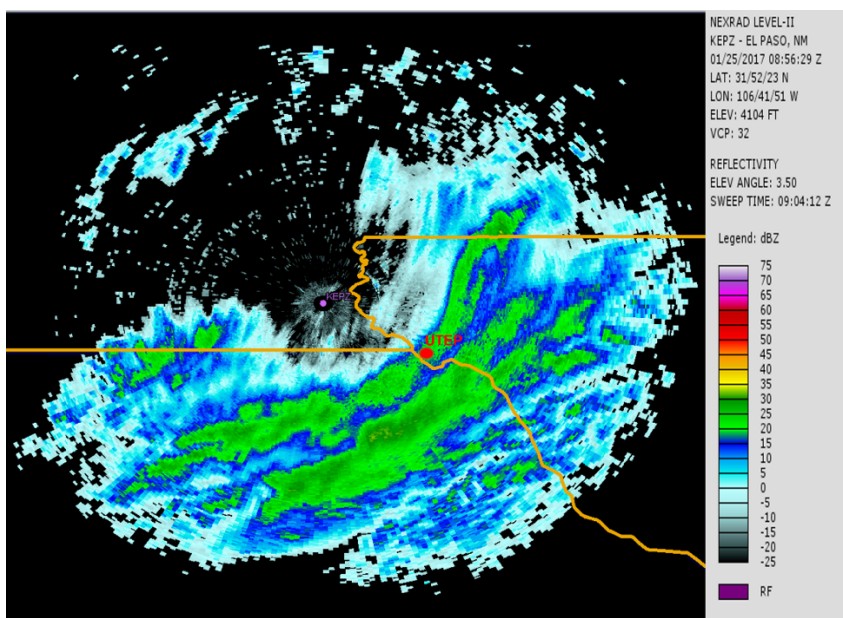

**Figure 2:** Base reflectivity imagery from the NEXRAD WSR-88D located at National Weather Service (NWS), Santa Teresa, New Mexico (purple dot). We can observe lighter precipitation at the research site UTEP shown in red dot. However, no precipitation is recorded at the ground station indicating occurrence of virga.


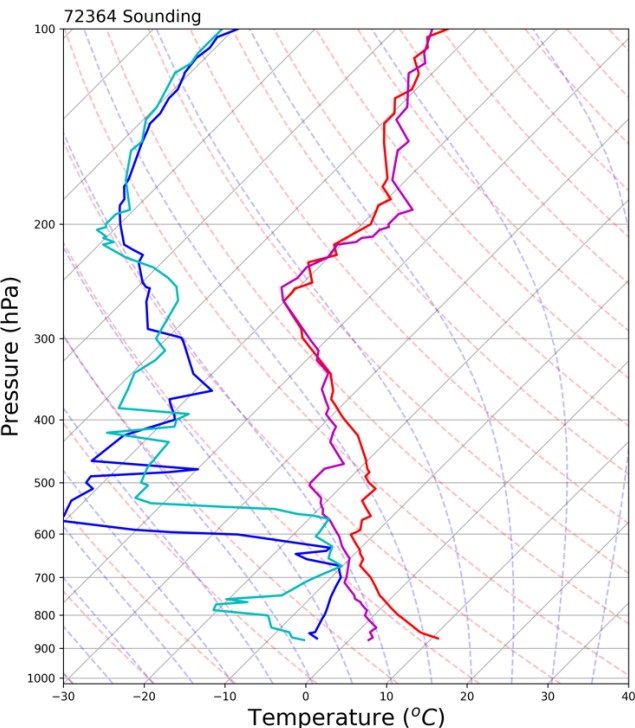



**Figure 3:** Skew-T plot for 25 January 2017, 00 UTC (T$_d$: blue, T: red) and 12 UTC (T$_d$: cyan, T: magenta) of the radiosonde data from the NWS, Santa Teresa, New Mexico.

The above case serves as an example of the steps followed in determining and confirming virga events during the study period of 2015-21. This criterion is not flawless but serves as a good starting point with the data availability for this region. All events which met the above criteria were visually inspected and ascertained as examples of virga. The next section will review the total virga found during the study and the search for patterns in their occurrences.

## 4 Regional seasonal virga patterns

When virga was classified by season, finer temporal resolution revealed important patterns. Different characteristics of virga, like any precipitation, are unique to each location, emphasizing the variability and evolution of humidity and dryness in the air. Seven years' worth of backscatter profiles from the ceilometer installed at UTEP campus were examined, and a long-term pattern of virga was established for this region. The ceilometer detected 50 virga events during the study period which met the detection criteria described in section 3. Figure 4 depicts the distribution of virga events throughout the study period (March 2015 – December 2021). The burgundy squares characterize days with virga, while the red squares represent days when the ceilometer data was unavailable. Figure 4 (a) illustrates that virga events occurred during the dry months of the year, with most of the events occurring during the winter and spring seasons. Some occurrences have also been reported in the early and late fall seasons.

Throughout the study, 2021 had the highest number of virga events. This high rate of successful virga detection can be attributed to two factors: a new ceilometer CL51 (which was commissioned in August 2020), with a greater vertical range and more powerful laser replacing the previous CL31, and the instrument's continuous operation. Most virga events in 2021 occurred during the winter and spring seasons, with only one occurring in the early fall. Three virga events were recorded in December 2020, with one event each in January, February, and April, and two in March. Similarly, to 2021, most of the virga events in 2019 occurred during the winter and spring months, with only one occurring in late December. 2015 saw the least number of virga events, followed by 2018 and 2017 due to the unavailability of the ceilometer dataset as a result of technical difficulties and maintenance issues. Even though continuous ceilometer data was available throughout the year, 2016 saw the fewest virga events.

Figure 4 (b) reveals a pattern in virga occurrences. Not a single virga event was recorded in May, June, July, and August. From June to October, El Paso has humid days when the average monthly precipitation ranges from 0.43 – 1.67 inches, as seen in Figure 4 (c). Even though September has an average of 1.52 inches of precipitation, three virga events were observed. On the remaining days of the year, this region is dry with plenty of sunlight (Karle et al., 2020). During the summer monsoon season, conditions are mostly humid and thus not ideal for virga occurrence. January recorded the most virga events, followed by March and April, which are some of the driest months of the year, with average precipitation of less than 0.4 inches. Not a single event was observed in October and only two were recorded in November.



240

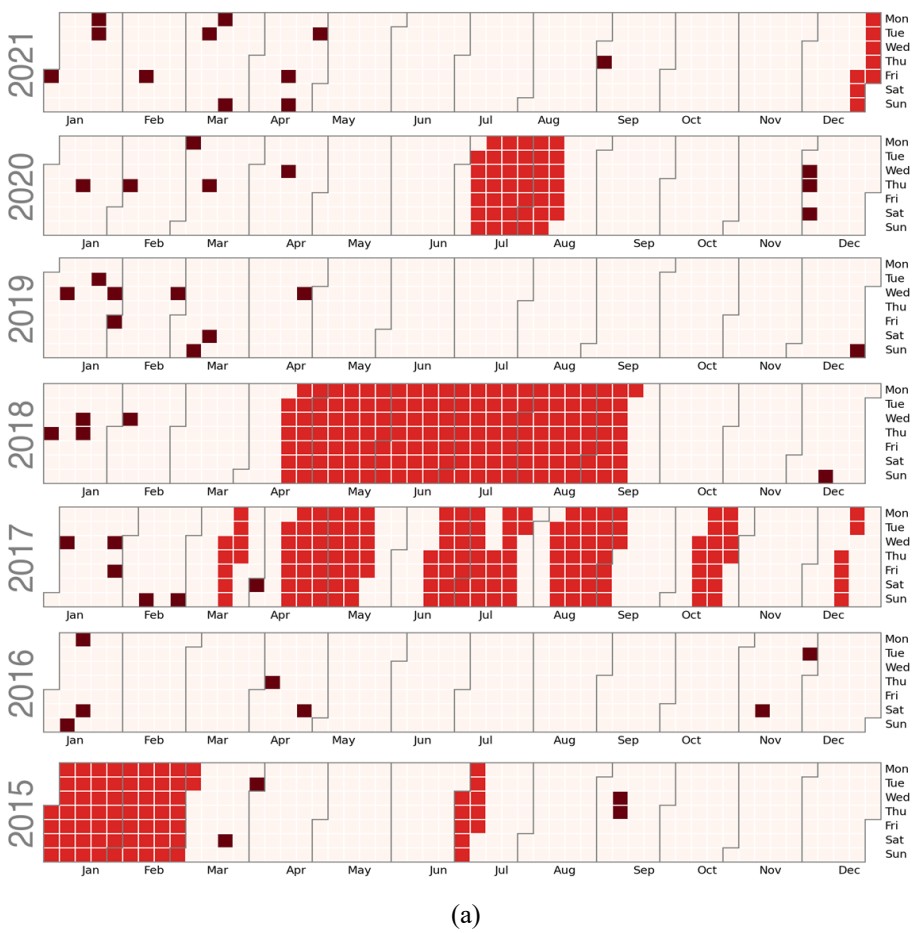

(a)

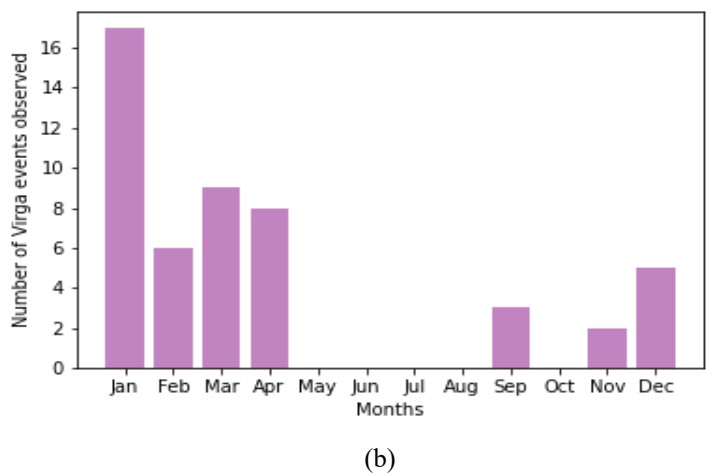

(b)





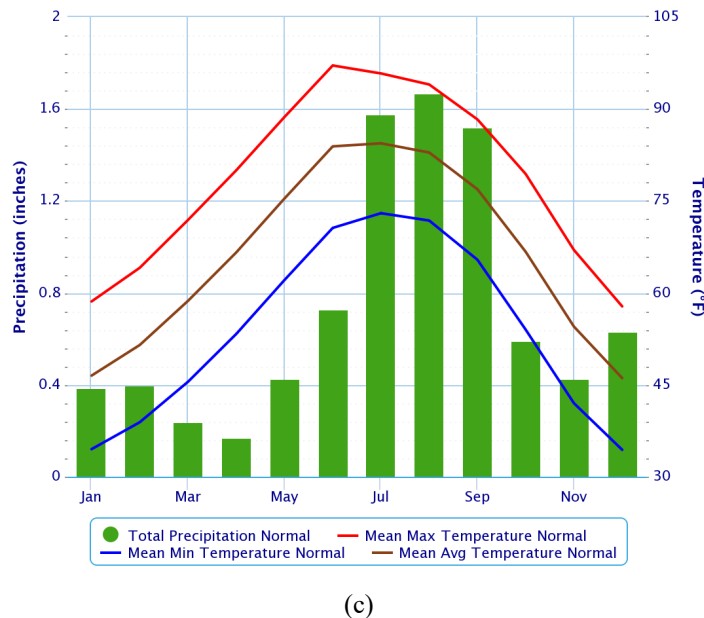

(c)

**Figure 4:** (a) A total of 50 virga events (burgundy) observed over the 7 years period (2015-2021). There are days when the instrument was not functional, such days are marked in red, (b) monthly distribution of the virga events, (c) monthly climate normal (1991-2020) for El Paso, Texas (Image Courtesy: National Weather Service, https://weather.gov)

## 5 Impact of virga on ground measurements

The observed virga events were classified into two types based on a thorough analysis of the aerosol backscatter profiles: columnar and non-columnar virga events. We observed a columnar aerosol profile from the base of the virga entering the surface layer in columnar occurrences using a ceilometer, whereas aerosols were not detected between the virga and the surface aerosol layer in non-columnar events. Twenty of the 50 events observed were columnar, while the remaining 30 were non-columnar. In the following section, we will examine two case studies representing each of the virga types mentioned above.

### 5.1 Case study 01: Columnar virga event of 31 March 2015

The virga event on March 31, 2015, was unusual in several ways. The virga occurred between 11 and 15 UTC and lasted for an hour. The occurrences, however, were not continuous, and we suspect the cloud moved away from the ceilometer, hence the discontinuity in virga detection. At 17 UTC, we witness another virga event, lasting more than 5 hours (till 22:45 UTC). The initial height of the virga (precipitation (red) depth from below the base of the cloud until it completely evaporates, i.e., no precipitation signal traced by the ceilometer) is slightly more than 0.5 km. However, as seen in Figure 5, the streak of aerosols (green) elongates below the precipitation and enters the surface aerosol layer (blue), forming a columnar structure. During the virga episode cloud base appears to be well above 4 km.





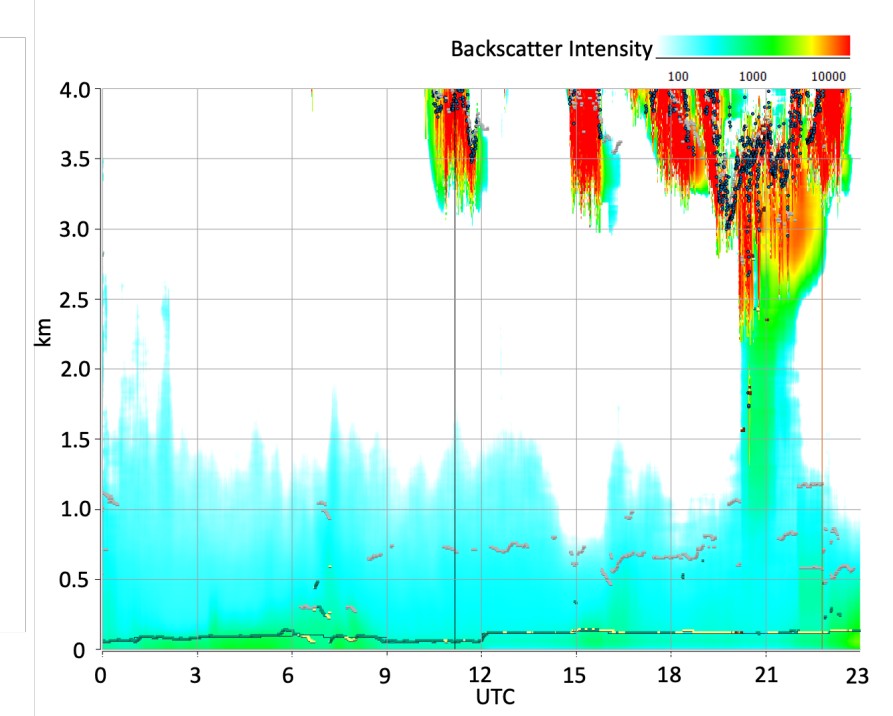


**Figure 5:** Virga event from 31 March 2015, when a columnar aerosol profile can be between 20-22 UTC reaching the surface aerosol layer.

Figure 6 (a, c, d) depicts hourly (in UTC) measurements from the CAMS 12 for March 31, 2015. According to (Theodore
Fujita, 1990) and (Wakimoto, 1985), a downburst (microbursts) is a strong downdraft that causes an outflow of intense winds
at or near the surface. Wind speed and maximum wind gust accelerated after 19 UTC and reached their maximum values (14.5
and 27.3 miles/hr, respectively) for the day between 21-22 UTC. However, since the ground measurements were hourly
averaged, the spike in wind gust due to dry microburst is missed out (Figure 6 a). The precipitation evaporated after descending
below the cloud base and into the dry layer, causing the air to cool and become negatively buoyant, as shown in Figure 5.
During the virga event, the evaporation or sublimation of melting particles is accompanied by the absorption of latent heat,
which eventually leads to a progressive cooling of the lower atmospheric layers especially below 1.5 km (Lolli et al., 2017).
Since the vertical profiles of temperature and dewpoint temperature in Figure 6 b exhibit a deep and dry layer, changes in wind
speed and maximum wind gust at 19 UTC can also be attributed to dry microbursts due to the availability of favourable
environmental conditions. On the other hand, temperatures show a steep decline during this period, indicating that the moisture
content in the air from the virga contributed to surface cooling (Figure 6 a). The relative humidity and dewpoint temperature
have a local peak at 12 UTC and 15 UTC, respectively, before falling precipitously as seen in Figure 6 c. At 19 UTC, the
relative humidity and dewpoint temperature rise dramatically, peaking at 23 UTC, thus providing another good characteristic
of the virga phenomenon detected by the ceilometer. In Figure 6 d, the particulate matter (PM) concentration charts exhibit





varying characteristics, at the beginning of the day between 0-4 UTC, both $PM_{2.5}$ and $PM_{10}$ display increase in concentration,

however at 19-22 UTC when the ceilometer observes the columnar structure from the virga, we see an collective increase in

the PM concentrations, especially at 22 UTC when we observe $PM_{2.5}$ (11.6 µg/m³) and $PM_{10}$ (81.4 µg/m³). We believe when

the virga produced dry microburst air hit the ground it spread out. The increase in the maximum wind gust was representing

the gust front which led to the increase in the PM concentrations by lifting the aerosols from the ground into the air. Another

possibility which could explain the increase in $PM_{2.5}$ levels is the cloud condensation nuclei (CCN) reaching the ground after

evaporation of the droplets.

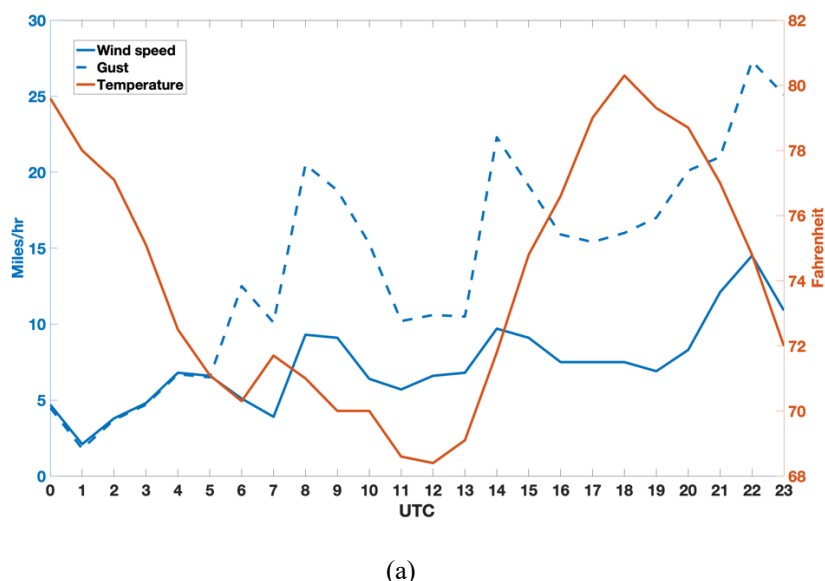

(a)





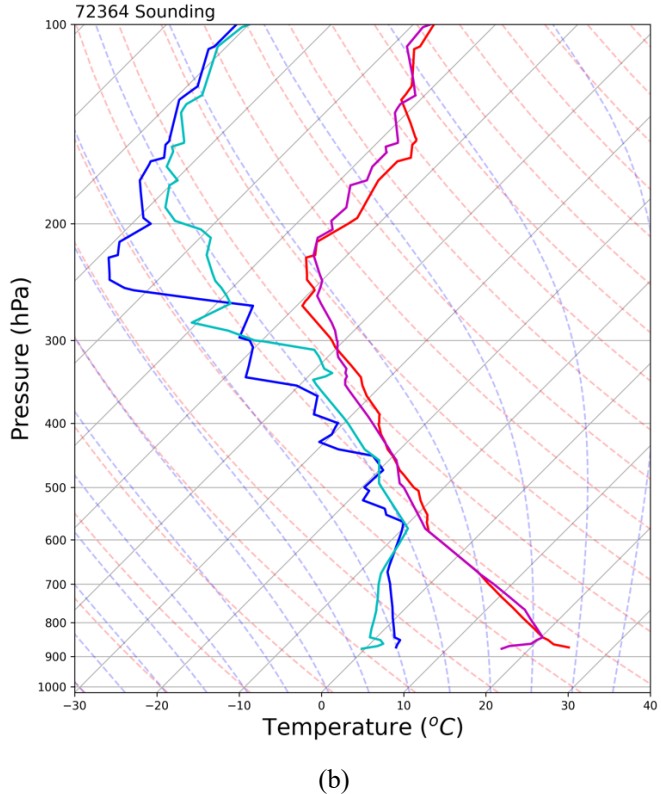


(b)

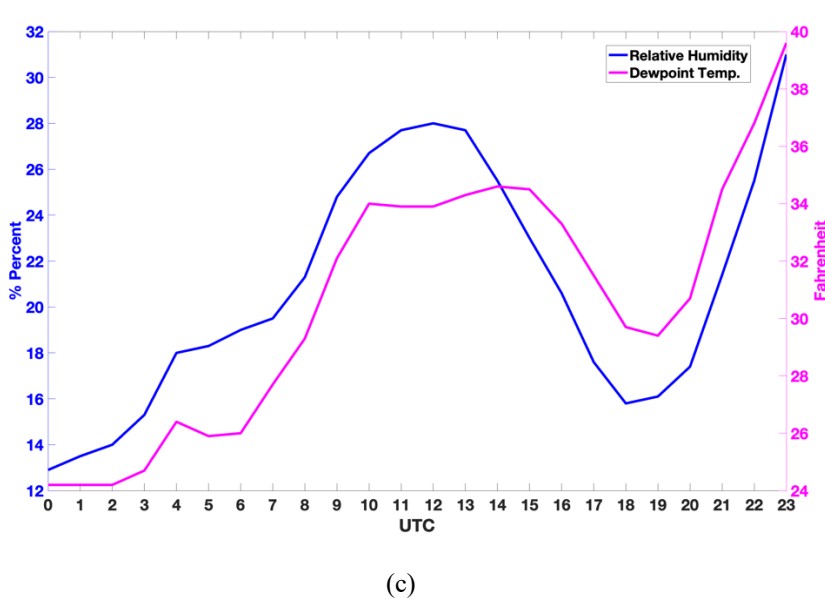

(c)



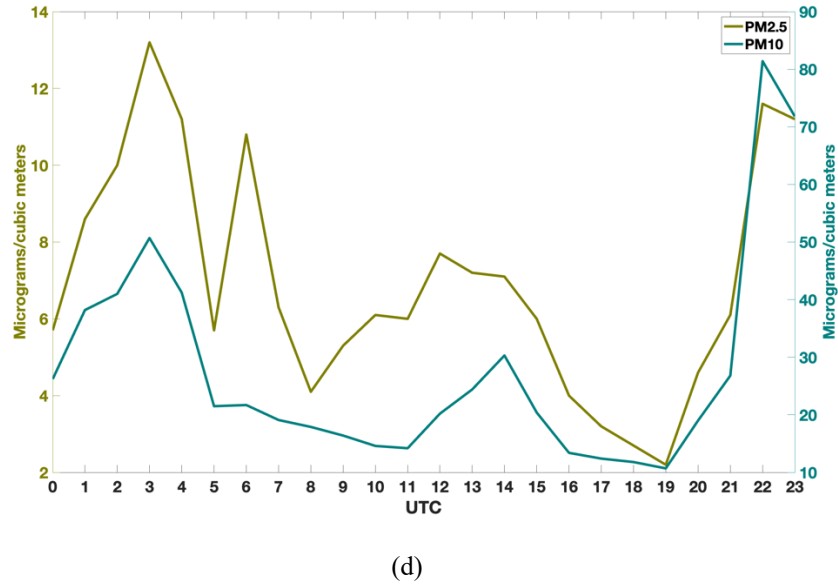

(d)

**Figure 6:** Hourly surface measurements from TCEQ CAMS 12 UTEP for 31 March 2015, (a) wind speed (solid blue), wind gust (dotted blue) in miles/hr and surface temperature (red) in Fahrenheit; (b) sounding from NWS for 31 March 2015 12 UTC ($T_d$: cyan, T: magenta) and 1 April 2015 00 UTC ($T_d$: blue, T: red); (c) relative humidity (purple) and dewpoint temperature (pink); (d) $PM_{2.5}$ concentration (olive) and $PM_{10}$ concentration (teal).

### 5.2 Case study 2: Virga event of 10 March 2019

On March 10, 2019, the ceilometer recorded a 16-hour-long virga event beginning at 4 CST (+1 MST) and ending at 20 CST, followed by precipitation. Figure 7 depicts the CL31 ceilometer's attenuated aerosol backscattering. The cloud base height is roughly 4 km. Again, the ceilometer cloud detection algorithm fails to calculate cloud base height accurately and misidentifies signals from larger rain droplets as clouds. The virga's height ranges between 1-1.5 km from 4-18 CST and gradually decreases. From 4 to 18 CST, the virga's height ranges between 1-1.5 km and gradually decreases. We observe some precipitation reaching the ground around 21 CST, but the ground instrument (rain gauge) fails to capture this relatively little precipitation. The red profile indicates a strong backscatter signal indicating rain, while the yellow and green profiles indicate reduced intensities. At 9 CST, we see a small section of virga entering the surface layer, but it is not as noticeable as in Case 1 with the columnar section. During 4-9 CST and 9:30-16 CST, no aerosol signal is recorded between the virga and the surface layer, indicating a disconnect between the virga and the surface aerosol layer.





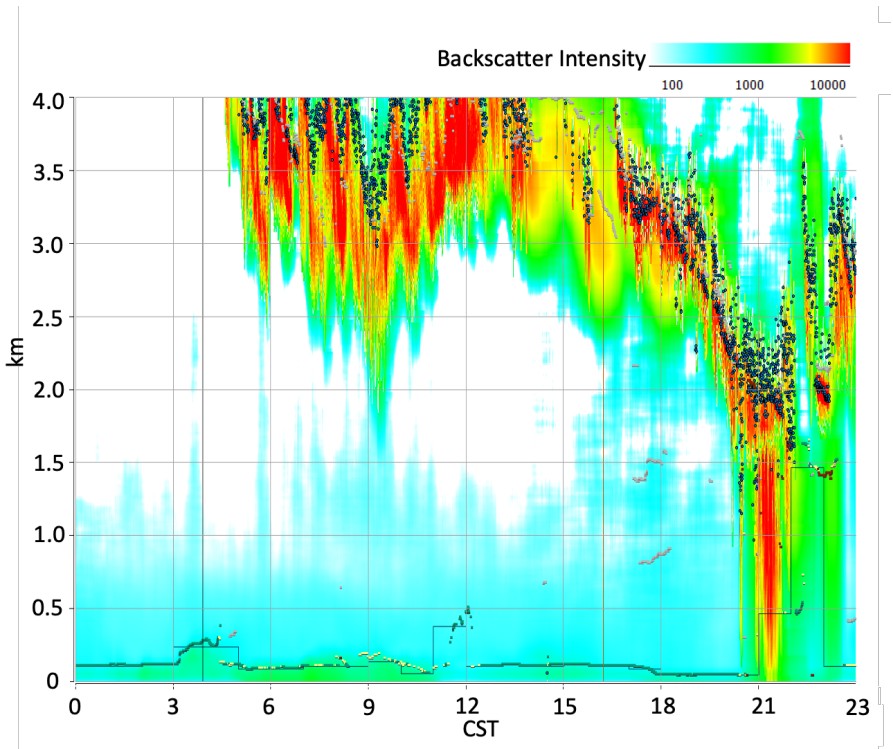

**Figure 7:** Virga event from 10 March 2019, starting from 06-19 CST, light precipitation can also be observed around 21 CST.


Maximum wind gusts were higher from 7 CST to 15 CST, as shown in (Figure 8 a). The temperature gradually rose after sunrise at 6 CST and peaked at 12 CST before sharply declining. We attribute the sudden drop in temperature to a significant increase in air humidity (Figure 8 c). Sounding profiles show the presence of dry air near and above the surface, especially at 00 UTC, because the difference between $T_d$ and T is large at the surface and peaks at 677 mb (Figure 8 b). The 12 UTC profile

is no different; dry air can be seen at the surface, even though the difference between $T_d$ and T is not as large as in 00 UTC. The air aloft appears to be moister as the difference between $T_d$ and T shrinks, and they get closer at 605 mb. The moisture in the upper atmosphere can be ascribed to the virga. The sudden huge increase in maximum wind gust after 6 CST is another clear indication of virga induced dry microburst. However, unlike in case 1, no significant fluctuations in PM concentrations were observed, even though maximum wind gusts were recorded during the virga event (Figure 8 d). This leads us to believe

that no aerosol loading occurred in the surface layer following precipitation evaporation.



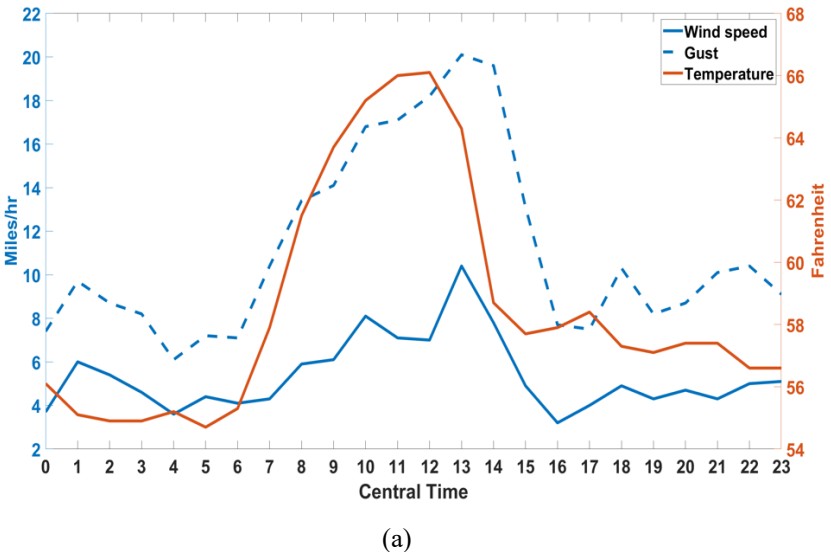

(a)


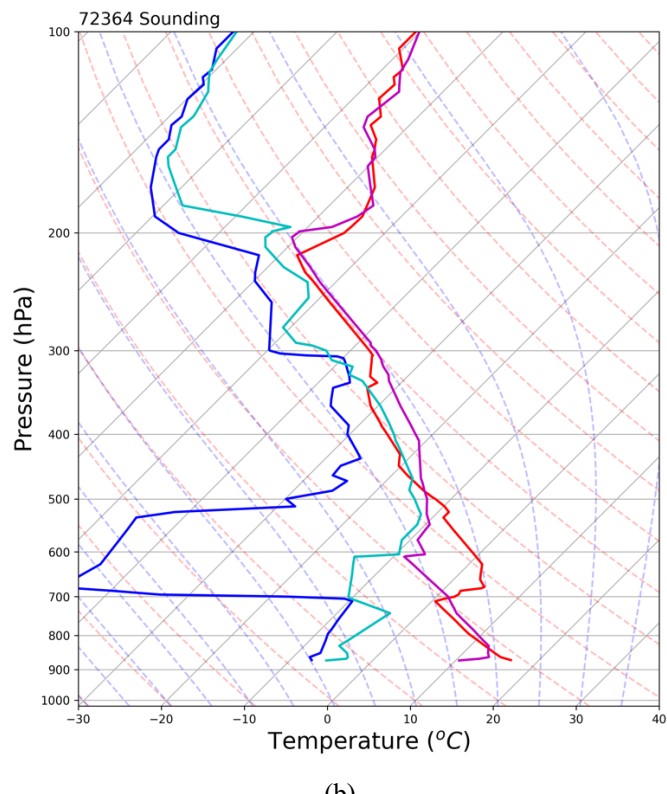

(b)




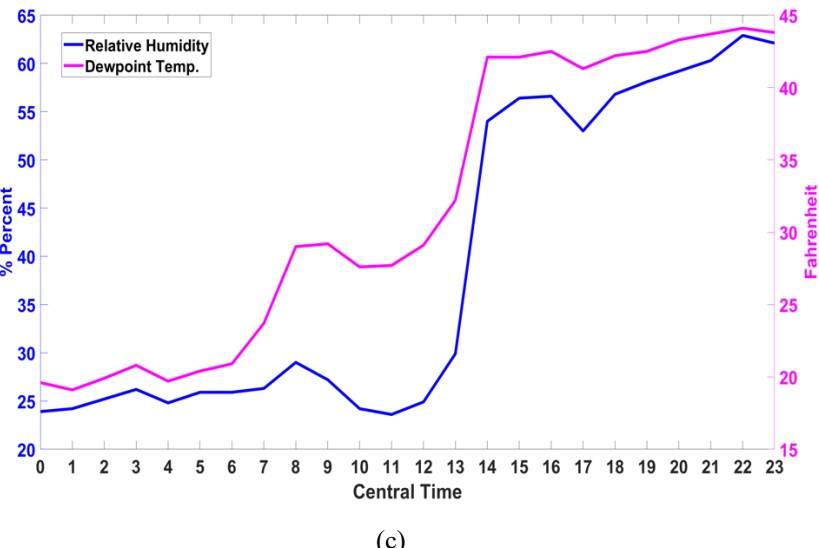

(c)

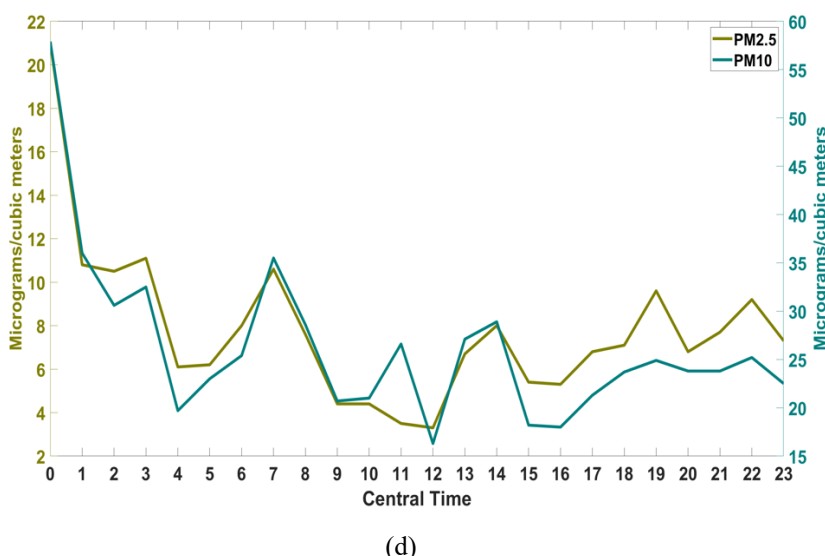

(d)

**Figure 8:** Hourly surface measurements (CST) from TCEQ CAMS 12 UTEP for 10 March 2019, (a) wind speed (solid blue), wind gust
(dotted blue) in miles/hr and surface temperature (red) in Fahrenheit; (b) sounding from NWS for 10 March 2019 at 00 UTC ($T_d$: blue, T:
red) and 12 UTC ($T_d$: cyan, T: magenta); (c) relative humidity (purple) and dewpoint temperature (pink); (d) $PM_{2.5}$ concentration (olive) and
$PM_{10}$ concentration (teal).




## 5.3 Relationship between ground measurements during virga events

We performed linear regression for several ground measurements during the virga events to quantify and analyse changes at the ground level. The rate of change of wind speed ($\Delta$WS) during the episode was compared with rate of change of $\Delta PM_{2.5}$, $\Delta PM_{10}$, $\Delta T_{air}$, and $\Delta T_d$ during both the columnar and non-columnar virga events as seen in (Table 1). We calculated p-value, a statistical parameter to validate a hypothesis against observed data and observed that all P values were higher than 0.05. Which means we cannot affirm that the slope calculated for the linear regression is not zero with 95% confidence interval. The R-squared values show slight improvement for most of the regressions for column virga cases. $\Delta$WS vs $\Delta PM_{2.5}$ had higher R-squared value in columnar episodes than in the other cases. All of this indicate that the hourly surface values which were obtained from the CAMS hinders the detection of the virga as well as microburst effects on the surface measurement time series.

**Table 1:** Regression parameters for several surface parameters of the Column virga cases (c) and not column virga cases (nc). WS = wind speed, $PM_{2.5}$ = particulate matter 2.5 μm, $PM_{10}$ = particulate matter 10 μm, $T_{air}$ = air temperature, $T_d$ = dew point temperature. Linear fit parameters: $R^2$ = squared correlation coefficient, P = p-value, N = sample size. Values in parenthesis are the standard error.

| Regression | Intercept | Slope | $R^2$ | P | N |
|---|---|---|---|---|---|
| $\Delta$WS vs $\Delta PM_{2.5}$ (nc) | -0.7 (1.3) | 0.05 (0.49) | 0.0004 | 0.92 | 28 |
| $\Delta$WS vs $\Delta PM_{2.5}$ (c) | -1.1 (3.0) | 1.7 (1.1) | 0.11 | 0.15 | 18 |
| $\Delta$WS vs $\Delta PM_{10}$ (nc) | 0.8 (1.9) | -0.34 (0.72) | 0.008 | 0.63 | 28 |
| $\Delta$WS vs $\Delta PM_{10}$ (c) | 11.2 (5.6) | -0.6 (2.2) | 0.004 | 0.78 | 16 |
| $\Delta$WS vs $\Delta T_{air}$ (nc) | -0.1 (0.4) | 0.01 (0.16) | 0.004 | 0.93 | 28 |
| $\Delta$WS vs $\Delta T_{air}$ (c) | 0.51(0.69) | 0.27 (0.26) | 0.05 | 0.31 | 18 |
| $\Delta$WS vs $\Delta T_d$ (nc) | 0.12 (0.79) | -0.32 (0.33) | 0.03 | 0.34 | 27 |
| $\Delta$WS vs $\Delta T_d$ (c) | 1.50 (0.78) | -0.27 (0.31) | 0.045 | 0.38 | 17 |

## 6 Discussion and Conclusions

This study investigated and characterized virga events in El Paso, Texas, using a combination of ceilometer, radiosonde, radar, meteorological and PM ground measurements and analysed its impact on local air quality. In the literature, very few studies





cover a wide range of virga forms and circumstances. However, this work has special merit due to the many events detected, classified, and analysed. We significantly extended our research and investigated the virga's impact on ground level PM concentrations. A gradient in attenuated aerosol backscatter intensities detected by ceilometers (Cl31 and CL51) helped identify virga. Retrieving the cloud base height during some virga events was difficult due to large rain droplets. Backscatter

profiles from 2015 to 2021 revealed 50 virga events that occurred locally. The inspection of virga events during the study period revealed a seasonal pattern of occurrence. No virga event was observed during the summer season when the moisture content in the region is high due to the arrival of the American monsoon. It was further shown that January reported maximum virga events followed by March.

Every event observed and analysed during this work was unique in its way. No two events were the same; however, we

identified and categorized these 50 events into two types: columnar and non-columnar events. Columnar virga events displayed a columnlike aerosol structure below the virga entering the surface layer, whereas non-columnar events displayed no such structures, but instead, we observe the absence of aerosol between the virga and surface layer. Since it was impossible to discuss all 50 events in this paper, we presented only the most significant ones as case studies. The virga cases discussed in this work lasted for more than 4 hours, during which no precipitation was recorded at the ground level, however the surface

measurements did indicate an increase in moisture content in the air. Both the events were characterized with gusty winds as indicated by the hourly maximum wind gusts data. Non-columnar virga case of 10 March 2019 was gustier than the columnar event. Relative humidity and dewpoint temperature data in both the cases showed a sudden increase during the virga phase indicating a sudden increase in the moisture content in the air. The sudden increase in PM concentrations in the columnar virga episode on March 10, 2019, can be attributed to a microburst-produced gust front, which caused a sudden spike in PM levels

by lifting aerosols from the ground into the air. Another possible explanation for the sudden increase in $PM_{2.5}$ levels is cloud condensation nuclei (CCN) reaching the ground after droplet evaporation.

This study also shows that a ceilometer can be a good and affordable alternative to lidars for detection of virga episodes and could provide more information on dry microbursts. Ceilometers are available throughout all the airports in the United States, but they gather mostly cloud height information. If the attenuated aerosol backscatter profiles from these ceilometers are made

available, then they can provide useful information on phenomena such as virga and dry microbursts.

Further studies exploring the remaining interesting instances (48 virga cases that were not discussed in this article) will be intriguing and required to have a deeper insight into the mechanisms and effects of virga on air quality. A study like this would allow researchers to investigate the possibility of a strong connection between vertical winds, virga, and a local rise in PM levels. It also emphasizes the importance of having diverse instrumentation at the El Paso site, such as sonic anemometers,

wind profilers, and barometers, which will provide a comprehensive dataset that will further enhance our understanding of virga and dry microbursts in the region. This research work will undoubtedly be a starting point for researchers to better comprehend the link between virga events and air quality. It will be worth analysing the impact virga has on the climatology of precipitation in the semi-arid region.




**Data availability**

Data and additional graphs related to this article are available upon request to the corresponding author.

**Author contributions**

The conceptualisation was performed by NK, RS, RF and WS. FM provided the ground measurements and NK, and RS curated the data. Methodology and investigation were completed by NK and RS. NK, RS, RF, and WS performed the analysis. NK drafted the original manuscript, which was reviewed and edited by RF and WS. CI contributed significantly to improving the

manuscript. Supervision was the responsibility of RS and WS, and RF acquired funding for the project.

**Competing interests**

The corresponding author nor the co-authors declare that they have no conflict of interests.

**Acknowledgements**

The authors want to specially thank Belay Demoz for providing useful research suggestions pertaining to virga. We are thankful

to the Texas Commission on Environmental Quality (TCEQ) and the National Weather Service (NWS) for supporting the data collection and archiving. Special thanks to NOAA Cooperative Science Centre in Atmospheric Sciences and Meteorology (NCAS-M) for the continuous support.

**Financial support**

This research has been supported by the NOAA/Educational Partnership Program under Cooperative Agreement

#NA16SEC4810006 and the NOAA Cooperative Science Centre in Atmospheric Sciences and Meteorology (NCAS-M) and by the Texas Commission on Environmental Quality (TCEQ) funding.

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
