# Peer review of "Systematic analysis of virga and its impact on surface particulate matter observations"

_EGUsphere, 2022_

## Referee Comment (RC1)

AMT 2022-906 **Full Referee Report**

This paper uses ceilometer data to identify potential cases of virga in El Paso, TX.  It looks at the impact of virga on PM2.5 concentrations, where certain (columnar virga) cases at times led to surface enhancements of PM2.5 while others (non-columnar virga) did not.

The authors' response to initial reviewer feedback updated sections related to Case 1 (lines 265-290; columnar virga event with a dry microburst) and Case 2 (lines 320-345; non-columnar virga event) of the originally submitted manuscript.  Both of those updates, including their updated figures, provide clarity and improve the manuscript.

Given the relatively brief time scales involved in the outflow of microbursts, the statistical analysis of the influence on PM2.5 by potential virga-induced microbursts can be strengthened by using 5-minute averaged data from the continuous air monitoring stations network rather than hourly averaged data.

1. Does the paper address relevant scientific questions within the scope of AMT?
   Yes

2. Does the paper present novel concepts, ideas, tools, or data?
   Yes

3. Are substantial conclusions reached?
   For the two highlighted case study examples, yes.  For the analysis of the 50 cases over the 7 year period, using higher temporal resolution surface data would help substantiate possible correlations between wind gusts resulting from virga-induced dry microbursts and enhanced PM2.5 concentrations.

4. Are the scientific methods and assumptions valid and clearly outlined?
   The methods are clearly outlined.

5. Are the results sufficient to support the interpretations and conclusions?
   In general, yes. There is a note described below that

6. Is the description of experiments and calculations sufficiently complete and precise to allow their reproduction by fellow scientists (traceability of results)?
   Yes, enough of a description is provided to allow other researchers to carry out similar studies.

7. Do the authors give proper credit to related work and clearly indicate their own new/original contribution?
Yes

8. Does the title clearly reflect the contents of the paper?
Yes

9. Does the abstract provide a concise and complete summary?
Yes

10. Is the overall presentation well structured and clear?
Yes, particularly with the updated portions provided in the authors' response to initial reviewer feedback.

11. Is the language fluent and precise?
With minor suggested edits (provided) and the improvements in the descriptions of updated sections in the authors' response to initial reviewer feedback, it is on the right track.

12. Are mathematical formulae, symbols, abbreviations, and units correctly defined and used?
In the updated versions of Figure 6b and Figure 8b provided in the authors' response to reviewers, please use m/s for the wind speed and max gust instead of miles/hr. Figures 6a, 6c and 8a, 8c, use Fahrenheit whereas the Skew-T plots (Figures 6b and 8b) use Celcius. Suggest using Celsius for all.

13. Should any parts of the paper (text, formulae, figures, tables) be clarified, reduced, combined, or eliminated?
Aside from the minor note above regarding units, the updated versions of Figure 6 and Figure 8 provided by the authors' response to initial feedback address this.

14. Are the number and quality of references appropriate?
Yes

15. Is the amount and quality of supplementary material appropriate?
N/A

Comments and Suggestions:

1. Line 150: "a few miles away from the study site": clearly specify the distance in kilometers and the directional heading. CAMS 12 is later mentioned with a lat/lon provided, would make sense to do the same for the NWS Santa Teresa sounding site location.

2. This paper provides a way of identifying virga events using ceilometer data. Confidence in proper identification of virga events will be higher for some cases than others. What suggestions do the authors have for other researchers to further confirm virga events?  This is just a suggestion for consideration and the authors should not feel compelled to include this: The authors' could add suggestions for future researchers to help identify such cases and their influence on PM2.5 surface concentrations, such as (1) having an all-sky camera on the roof that saves one picture each minute, (2) releasing collocated radiosondes on demand during such events, and (3) obtaining higher temporal resolution data from surface monitors. Granted, this would require some additional funding but could provide things to consider for those who perform similar studies.

3. Using 5-minute instead of hourly data for the analysis in Table 1 was suggested in the initial review response and the authors responded:

   The 50 virga episodes that were classified in this work are unevenly distributed throughout the study period of 7 years. Since the 5-minute data was not available to us we decided to use the one-hour average data which is readily available in the public domain. We are grateful for your suggestion and will seek the 5-minute data for future use.

   The authors' response is understandable but it would be a missed opportunity to further improve this manuscript. The dry microburst events are short-lived and the 5-minute data may strengthen the analysis.

Minor suggested edits that do not change authors' intended meaning:

Line 30: "before even reaching" → "before reaching"

Line 35: "reviewed earth's climate system" → "reviewed Earth's climate system"

Lines 47-48: "radar and lidar, both ground-based, airborne, and satellite observations" please rewrite for clarity

Line 48: "(Wang et al., 2018)" → "Wang et al. (2018)"

Lines 51-52: "(Saikranthi et al., 2014)" → "Saikranthi et al. (2014)"

Lines 53-54: "(Airey et al., 2021)" → "Airey et al. (2021)"

Line 70: "(Jullien et al., 2020)" → "Jullien et al. (2020)"

Link 71: "(Grazioli et al., 2017)" → "Grazioli et al. (2017)"

Line 74: Use the degree symbol (in Word, go to 'Insert' and then 'Symbol') rather than a superscripted "0"

Line 79: "(Tost et al., 2016)" → "Tost et al. (2016)"

Line 94: Change the semicolon to a period (or could use semicolons throughout that list of the different section topics).

Line 100: "doppler" → "Doppler"

Line 102: Use the degree symbol rather than a superscripted "0"

Line 104: Delete "urban"

Line 105: "Sun city's" → "Sun City's"

Line 113: "aerosol layer height which can" → "aerosol layer height, which can"

Line 121: "CL31 was installed" → "A CL31 was installed"

Line 122: "CL51 has a" → "The CL51 has a"

Lines 135-136: "planetary boundary layer heights (Schafer et al. 2004) from the measured attenuated backscatter profiles." → "planetary boundary layer heights from the measured attenuated backscatter profiles (Schafer et al. 2004)."

Line 137: "for detection" → "for the detection"

Line 140: "doppler" → "Doppler"

Line 142: "diamter" → "diameter"

Figure 2 caption: "We can observe lighter precipitation at the research site UTEP shown in red dot." → "We can observe light precipitation at the research site (UTEP) shown by the red dot."

Line 216: "years' " → "years"  (no apostrophe)

Line 216: "at UTEP" → "at the UTEP"

Line 218: "section 3" → "Section 3"

Line 227: "Similarly, to 2021, most" → "Similar to 2021, most"

Lines 228-229: "2015 saw the least number of virga events" → "2015 had the fewest virga events"

Figure 4 caption is missing a period at the end of it.

Line 256: "Case study 01" → "Case study 1"

Line 259: (till 22:45 UTC) → "(until 22:45 UTC)"

Line 263: "During the virga episode cloud base appears to be well above 4 km." → "During the virga episode, the cloud base appears to be well above 4 km."

Line 312: "ground around 21 CST" → "ground at approximately 21 CST" (and from the authors' response to reviewers the times throughout may end up in UTC)

Line 354: "(Table 1)" → "Table 1"

Line 354: "calculated p-value" → "calculated the p-value"

Line 355: "P values" → "p-values"

Lines 355-356: "higher than 0.05. Which means" → "higher than 0.05; meaning"

Line 357: "vs" → "vs."

Lines 357-358: "had higher R-squared value in" → "had a higher R-squared value in"

Line 358: "indicate" → "indicates"

Lines 372-373: "We significantly extended our research and investigated the virga's impact on ground level PM concentrations." → "Our research extends the literature by providing an initial investigation into virga's impact on ground level PM concentrations." (or something to that effect)

Line 373: "Cl31" → "CL31"

Line 389: "March 10, 2019," → "10 March 2019" (for consistency in how dates are formatted and is something to check throughout, most have DD Month YYYY)

Line 403: "in the semi-arid region." → "in semi-arid regions."

Minor suggested edits the authors' response to initial reviewer feedback:

Figure 5 caption: "around 20-22 UTC" → "approximately 20-22 UTC" or "~20-22 UTC"

End of Page 5: "virga in case 1 was" → "virga in Case 1 was"

Last page: "(figure 8b)" → "(Figure 8b)"; "figure 8 c" → "Figure 8c"; "figure 8d" → "Figure 8d"

---

## Author Comment (AC2)

Authors response to AMT 2022-906 Anonymous Referee #2 Report

We would like to express our gratitude to the referee for taking the time to review our first response. Your comments and suggestions have been invaluable in helping us improve the quality and clarity of our work. The referee comments/suggestions are in black, and our responses are in red; the listed line numbers refer to the lines of the original manuscript where the new corrections are observed.

This paper uses ceilometer data to identify potential cases of virga in El Paso, TX. It looks at the impact of virga on PM2.5 concentrations, where certain (columnar virga) cases at times led to surface enhancements of PM2.5 while others (non-columnar virga) did not.

The authors' response to initial reviewer feedback updated sections related to Case 1 (lines 265-290; columnar virga event with a dry microburst) and Case 2 (lines 320-345; non-columnar virga event) of the originally submitted manuscript. Both of those updates, including their updated figures, provide clarity, and improve the manuscript.

Thank you for taking the time to review our response to your first review report.

Given the relatively brief time scales involved in the outflow of microbursts, the statistical analysis of the influence on PM2.5 by potential virga-induced microbursts can be strengthened by using 5-minute averaged data from the continuous air monitoring stations network rather than hourly averaged data.

We agree with the referee, however, we tried hard and were unable to get access to the finer resolution dataset.

1. Does the paper address relevant scientific questions within the scope of AMT?
Yes

2. Does the paper present novel concepts, ideas, tools, or data?
Yes

3. Are substantial conclusions reached?
For the two highlighted case study examples, yes. For the analysis of the 50 cases over the 7 year period, using higher temporal resolution surface data would help substantiate possible correlations between wind gusts resulting from virga-induced dry microbursts and enhanced PM2.5 concentrations.

4. Are the scientific methods and assumptions valid and clearly outlined?
The methods are clearly outlined.

5. Are the results sufficient to support the interpretations and conclusions?

In general, yes. There is a note described below that

6. Is the description of experiments and calculations sufficiently complete and precise to allow their reproduction by fellow scientists (traceability of results)?
Yes, enough of a description is provided to allow other researchers to carry out similar studies.

7. Do the authors give proper credit to related work and clearly indicate their own new/original contribution?
Yes

8. Does the title clearly reflect the contents of the paper?
Yes

9. Does the abstract provide a concise and complete summary?
Yes

10. Is the overall presentation well structured and clear?
Yes, particularly with the updated portions provided in the authors' response to initial reviewer feedback.

11. Is the language fluent and precise?
With minor suggested edits (provided) and the improvements in the descriptions of updated sections in the authors' response to initial reviewer feedback, it is on the right track.

12. Are mathematical formulae, symbols, abbreviations, and units correctly defined and used?
In the updated versions of Figure 6b and Figure 8b provided in the authors' response to reviewers, please use m/s for the wind speed and max gust instead of miles/hr. Figures 6a, 6c and 8a, 8c, use Fahrenheit whereas the Skew-T plots (Figures 6b and 8b) use Celcius. Suggest using Celsius for all.

Thank you for the suggestions. We have revised the above-mentioned figures with appropriate units to address the referee suggestions.

13. Should any parts of the paper (text, formulae, figures, tables) be clarified, reduced, combined, or eliminated?
Aside from the minor note above regarding units, the updated versions of Figure 6 and Figure 8 provided by the authors' response to initial feedback address this.

14. Are the number and quality of references appropriate?
Yes

15. Is the amount and quality of supplementary material appropriate?

N/A
Comments and Suggestions:
1. Line 150: "a few miles away from the study site": clearly specify the distance in kilometers and the directional heading. CAMS 12 is later mentioned with a lat/lon provided, would make sense to do the same for the NWS Santa Teresa sounding site location.

The following revision is made to the manuscript:
Radiosonde observations were obtained from the nearest National Weather Service (NWS) (31°52'33" N, 106°36'39" W) located at Santa Teresa, New Mexico, 21 km away from the study site.

2. This paper provides a way of identifying virga events using ceilometer data. Confidence in proper identification of virga events will be higher for some cases than others. What suggestions do the authors have for other researchers to further confirm virga events? This is just a suggestion for consideration and the authors should not feel compelled to include this: The authors' could add suggestions for future researchers to help identify such cases and their influence on PM2.5 surface concentrations, such as (1) having an all-sky camera on the roof that saves one picture each minute, (2) releasing collocated radiosondes on demand during such events, and (3) obtaining higher temporal resolution data from surface monitors. Granted, this would require some additional funding but could provide things to consider for those who perform similar studies.

Thank you for the excellent suggestions. We do appreciate it. The last paragraph which describes the future scope of the work is revised to include the suggestions from the referee.

The availability of higher temporal resolution ground measurements will undoubtedly improve and solidify the correlation between the various parameters discussed in Table 1. If funds are available, using an all-sky camera (capable of capturing finer temporal resolution images) in conjunction with the ceilometer would greatly aid in capturing the virga precipitation. During the virga event, launching collocated radiosondes could provide an excellent dataset of vertical atmospheric profiles, especially the wind flows. A comprehensive study that includes such instrumentation and approaches would allow researchers to investigate the possibility of a strong connection between vertical winds, virga, and a local rise in PM levels. It also emphasizes the importance of having diverse instrumentation at the El Paso site, such as sonic anemometers, wind profilers, and barometers, which will provide a comprehensive dataset that will further enhance our understanding of virga and dry microbursts in the region. This research work will undoubtedly be a starting point for researchers to better comprehend the link between virga events and air quality. It will be worth analyzing the impact virga has on the climatology of precipitation in the semi-arid region.

3. Using 5-minute instead of hourly data for the analysis in Table 1 was suggested in the initial review response and the authors responded:

The 50 virga episodes that were classified in this work are unevenly distributed throughout the study period of 7 years. Since the 5-minute data was not available to us we decided to use the one-hour average data which is readily available in the public domain. We are grateful for your suggestion and will seek the 5-minute data for future use.

The authors' response is understandable, but it would be a missed opportunity to further improve this manuscript. The dry microburst events are short-lived and the 5-minute data may strengthen the analysis.

We agree with the referee, however, we tried hard and were unable to get access to the finer resolution dataset. Based on the availability, future work will include higher temporal resolution data.

Minor suggested edits that do not change authors' intended meaning:
Line 30: "before even reaching" → "before reaching"

The sentence has been revised.

Line 35: "reviewed earth's climate system" → "reviewed Earth's climate system"

The sentence has been revised.

Lines 47-48: "radar and lidar, both ground-based, airborne, and satellite observations" please rewrite for clarity

Previous studies investigated it using remote sensing instruments such as ground-based or airborne radar and/or lidar, and in some cases satellite observations.

Line 48: "(Wang et al., 2018)" → "Wang et al. (2018)"

The citation has been revised.

Lines 51-52: "(Saikranthi et al., 2014)" → "Saikranthi et al. (2014)"

The citation has been revised.

Lines 53-54: "(Airey et al., 2021)" → "Airey et al. (2021)"

The citation has been revised.

Line 70: "(Jullien et al., 2020)" → "Jullien et al. (2020)"

The citation has been revised.

Link 71: "(Grazioli et al., 2017)" → "Grazioli et al. (2017)"

The citation has been revised.

Line 74: Use the degree symbol (in Word, go to 'Insert' and then 'Symbol') rather than a superscripted "0"

Symbol revised

Line 79: "(Tost et al., 2016)" → "Tost et al. (2016)"

The citation has been revised.

Line 94: Change the semicolon to a period (or could use semicolons throughout that list of the different section topics).

Revision made

Line 100: "doppler" → "Doppler"

Revised

Line 102: Use the degree symbol rather than a superscripted "0"

Symbol revised

Line 104: Delete "urban"

Revised

Line 105: "Sun city's" → "Sun City's"

Revised

Line 113: "aerosol layer height which can" → "aerosol layer height, which can"

Addressed

Line 121: "CL31 was installed" → "A CL31 was installed"

Addressed

Line 122: "CL51 has a" → "The CL51 has a"

Addressed

Lines 135-136: "planetary boundary layer heights (Schafer et al. 2004) from the measured attenuated backscatter profiles." → "planetary boundary layer heights from the measured attenuated backscatter profiles (Schafer et al. 2004)."

Addressed

Line 137: "for detection" → "for the detection"

Addressed

Line 140: "doppler" → "Doppler"

Addressed

Line 142: "diamter" → "diameter"

Revised

Figure 2 caption: "We can observe lighter precipitation at the research site UTEP shown in red dot." → "We can observe light precipitation at the research site (UTEP) shown by the red dot."

Addressed

Line 216: "years' " → "years" (no apostrophe)

Addressed

Line 216: "at UTEP" → "at the UTEP"

Revised

Line 218: "section 3" → "Section 3"

Addressed

Line 227: "Similarly, to 2021, most" → "Similar to 2021, most"

Addressed

Lines 228-229: "2015 saw the least number of virga events" → "2015 had the fewest virga events"

Addressed

Figure 4 caption is missing a period at the end of it.

Addressed

Line 256: "Case study 01" → "Case study 1"

Addressed

Line 259: (till 22:45 UTC) → "(until 22:45 UTC)"

Addressed

Line 263: "During the virga episode cloud base appears to be well above 4 km." → "During the virga episode, the cloud base appears to be well above 4 km."

Addressed

Line 312: "ground around 21 CST" → "ground at approximately 21 CST" (and from the authors' response to reviewers the times throughout may end up in UTC)

Addressed

Line 354: "(Table 1)" → "Table 1"

Addressed

Line 354: "calculated p-value" → "calculated the p-value"

Revised

Line 355: "P values" → "p-values"

Addressed

Lines 355-356: "higher than 0.05. Which means" → "higher than 0.05; meaning"

Addressed

Line 357: "vs" → "vs."

Addressed

Lines 357-358: "had higher R-squared value in" → "had a higher R-squared value in"

Addressed

Line 358: "indicate" → "indicates"

Addressed

Lines 372-373: "We significantly extended our research and investigated the virga's impact on ground level PM concentrations." → "Our research extends the literature by providing an

initial investigation into virga's impact on ground level PM concentrations." (or something to that effect)

Revised

Line 373: "Cl31" → "CL31"

Addressed

Line 389: "March 10, 2019," → "10 March 2019" (for consistency in how dates are formatted and is something to check throughout, most have DD Month YYYY)

Modified

Line 403: "in the semi-arid region." → "in semi-arid regions."

Addressed

Minor suggested edits the authors' response to initial reviewer feedback:
Figure 5 caption: "around 20-22 UTC" → "approximately 20-22 UTC" or "~20-22 UTC"
End of Page 5: "virga in case 1 was" → "virga in Case 1 was"
Last page: "(figure 8b)" → "(Figure 8b)"; "figure 8 c" → "Figure 8c"; "figure 8d" → "Figure 8d"_

All the points have been addressed in the manuscript.

Once again, we appreciate your time and expertise.

---

## Author Response (AR1)

**Authors response to AMT 2022-906 Referees Comments**

Authors response to AMT 2022-906 Anonymous Referee #1 Report 1

We would like to thank the referee for taking time to review our manuscript. We greatly appreciate your insights and suggestions and have taken them into consideration in revising our work. The referee comments/suggestions are in black, and our responses are in red; the listed line numbers (bold font style) refer to the lines of the revised manuscript where the new corrections are observed.

This work by Karle et al. (Systematic analysis of virga and its impact on surface particulate matter observations) presented some very interesting results regarding the viga precipitation, which is rarely studied. I only have some minor comments before the work can be accepted.

General comments:

1. I did not quite understand the large picture between virga precipitation and aerosol. In the abstract and in the case studies, it is stated that "We observed that during some of the columnar virga events, surface PM levels displayed a sudden upward trend indicating aerosol loading in the surface layer after precipitation evaporation." Should it be the opposite? That is, during the precipitation virga process, PM level should be downward. That is, most of these PM are used as rain drop nuclei. Maybe I missed your point. Please explain.

This is an excellent point raised by the reviewer and we would like to provide the following clarification.

As part of a larger picture between the virga events and aerosols we are underlining the occurrence of dry microbursts that are usually associated with virga precipitation. (Fujita, 1981; Wakimoto, 1985) define dry microburst as convectively driven small downdrafts of less than 4 km in outflow diameter accompanied by little or no rain between the beginning and end of the intense wind gusts for a short period. The dry microbursts are frequently associated with virga precipitation.

The ceilometer not only successfully detected the virga precipitation, but it also observed the aerosol loading in the surface layer. Since we observed a substantial increase in the surface measured maximum wind gusts during the virga event, we attributed these horizontal winds to dry microburst. Based on these observations we conclude that the sudden increase in the surface PM level (lasting for only couple of hours) was due to wind gusts associated with dry microburst. We also observe that with the gradual increase in humidity levels, the PM levels eventually drop, since these PM would serve as the raindrop nuclei as rightly pointed out by the referee. More detailed explanation of the columnar case can be found in our response to referee #2 (https://doi.org/10.5194/egusphere-2022-906-AC1).

Fujita, T. T. (1981). Tornadoes and Downbursts in the Context of Generalized Planetary Scales, Journal of Atmospheric Sciences, 38(8), 1511-1534. https://journals.ametsoc.org/view/journals/atsc/38/8/1520 0469_1981_038_1511_taditc_2_0_co_2.xml

Wakimoto, R. M. (1985). Forecasting Dry Microburst Activity over the High Plains, Monthly Weather Review, 113(7), 1131-1143. https://journals.ametsoc.org/view/journals/mwre/113/7/1520-0493_1985_113_1131_fdmaot_2_0_co_2.xml

1. In your Table 1, the wind speed and PM, wind speed and DeltaT have very week correlation. Do you think this is an observation issue (that is, your observation does not have finer temporal resolution)? Please explain.

Another good observation from the referee. We agree that there is a week correlation between the PM and winds based on the ground measurements. It is also true that a finer temporal resolution of the ground measurements will be able to capture the fluctuations in the wind speeds and maximum wind gusts associated especially with dry microburst. However, finer resolution data was not available to us and hence we decided to use one-hour data which is readily available in the public domain for calculations as shown in Table 1.

**Lines 661-672** we have highlighted our response/justification to the referee's question/comment.

Minor comments:

Fig. 7 and Fig.5, The legend for the "backscatter intensity", I think the unit is "Z"? Can you please change it to dBZ (like in Fig. 2)

The Figure 1 **(Line 230)**, Figure 5 **(Line 310)** and Figure 7 **(Line 402)** are revised by adding legend "dBZ" for the "backscatter intensity" .

For the sounding profile (e.g., Fig. 6b), please add a legend for different color curves.

All the sounding Figures (Figure 3, **Line 240**; Figure 6b, **Line 355**; Figure 8b, **Line 570**) in the manuscript are revised.

We hope that these revisions address your concerns and fully demonstrate the significance and originality of our research.
Thank you again for your review, and we look forward to the opportunity to resubmit our manuscript for your consideration.
* * *
Authors response to AMT 2022-906 Initial Referee # 2 Report 1

We deeply appreciate the time and effort the initial referee dedicated to providing valuable feedback towards helping us improve the manuscript. We have been able to incorporate changes to reflect most of the suggestions provided by the referee in our manuscript.
The referee comments/suggestions are in black, and our responses are in red, the listed line numbers (bold font style) refer to the lines of the revised manuscript where the new corrections are observed.

This manuscript suggests an interesting approach of measuring virga and the potential connection of virga with particulate matter concentrations at the surface using ceilometers and in conjunction with other supporting measurements.

Thank you for your comment.

Table 1 and other results throughout the manuscript could potentially be strengthened using 5-minute TCEQ CAMS data rather than the hourly averaged data. The 5-minute CAMS data is available upon request from TCEQ.

The 50 virga episodes that were classified in this work are unevenly distributed throughout the study period of 7 years. Since the 5-minute data was not available to us we decided to use the one-hour average data which is readily available in the public domain. We are grateful for your suggestion and will seek the 5-minute data for future use.

Include wind data on the Skew-T plots.

We have included wind barbs to all the sounding Figures (Figure 3, **Line 240**; Figure 6b, **Line 355**; Figure 8b, **Line 570**) in the manuscript are revised.

An all-sky fisheye camera colocated with the ceilometer saving an image everyone or five minutes could help with the verification of virga events and instances of dry microbursts.

That is a very good suggestion. We will implement it in near future, based on the availability of funds.

It is not clear that a wind gust in the absence of surface precipitation during what may be a period of a ceilometer-verified virga event is an indication of a dry microburst. It is consistent but inconclusive.

Ceilometer provides the columnar aerosol backscatter profile and based on that it successfully identified virga precipitation. Dry microbursts are defined by (Fujita, 1981 and Wakimoto, 1985) as convectively driven small downdrafts of less than 4 km in outflow diameter accompanied by little or no rain between the beginning and the end of the intense winds for a short duration of time. They are commonly associated with virga or precipitation that evaporates before reaching the ground.

Since we observed a substantial increase in the surface measured maximum wind gusts during the virga event, we attributed these horizontal winds to dry microburst. Of course, we also agree that more sophisticated remote sensing instrumentation, such as radio wind profiler or wind lidars, at the site could be helpful in providing more information on vertical winds such as to capture the downbursts. Ours represent the first study of virga events for our region. It is our hope that the publication of this manuscript will create an incentive to pursue these types of studies, which are particularly important in severe dry regions, such as ours.

Fujita, T. T. (1981). Tornadoes and Downbursts in the Context of Generalized Planetary Scales, Journal of Atmospheric Sciences, 38(8), 1511–1534. Retrieved Dec 19, 2022, from
https://journals.ametsoc.org/view/journals/atsc/38/8/1520-0469_1981_038_1511_taditc_2_0_co_2.xml

Wakimoto, R. M. (1985). Forecasting Dry Microburst Activity over the High Plains, Monthly Weather Review, 113(7), 1131–1143. Retrieved Dec 19, 2022, from
https://journals.ametsoc.org/view/journals/mwre/113/7/1520-0493_1985_113_1131_fdmaot_2_0_co_2.xml

The non-columnar Case Study 2 potential virga event on 10 March 2019 needs further explanation on the timing and the lack of a surface level PM enhancement.

This is an excellent point raised by the reviewer. We would like to use both the cases discussed in this paper to present a convincing justification. In addition, the following modification and explanation are added to the respective sections of the manuscript as well.

For uniformity and convenience of our readers, we have all the figures in UTC and figures 6 and 8 (a), (c) and (d) have a uniform y-axis scale respectively. Additionally, we have added markers for radiosonde data availability (dashed lines) and highlighted portion to emphasize the section of the figure under examination. We hope this will help convey our explanation to the reviewers and readers more effectively.

**Line 310:** Figure 5 revised and the region of interest is highlighted for the interest of the readers.

**Line 378-389**: We have provided more explanation of Case 1, the columnar virga event by adding the following paragraph.

Figure 6a shows an increase in surface wind speed and maximum gust during the virga event (shaded). The daytime temperature peaks at 19 UTC and then begins to fall. The radiosonde vertical profiles on 31 March 2015 at 12 UTC and 1 April 2015 at 0 UTC provide a better understanding of the thermodynamic state of the atmosphere within and above the boundary layer (Figure 6b). Temperature (red) and dew point temperature (blue) are further apart between 876 mb (1.3 km above sea level, a.s.l.) and 577 mb (4.7 km a.s.l.), indicating lower relative humidity at these levels. In the 0 UTC sounding, we observe the temperature drifting from the adiabatic ascent curve showing a diabatic behaviour. This loss of latent heat energy of the air parcel can be attributed to the air column which has cooled down after the evaporation of precipitation underneath the clouds. Another important piece of information we obtain from the soundings is the wind intensity near the surface. As seen in the wind barbs associated with the 12 and 0 UTC sounding, winds are calm between 732 mb (2.7 km a.s.l.) and 655 mb (3.7 km a.s.l.). However, near the surface we observe strong winds (around 10-11 m/s) which appears to be decoupled from winds between the cloud base and the surface. Based on the thermodynamic and meteorological evidence, it can be concluded that virga in Case 1 was intense in nature.

**Line 448:** Figure 7 revised and the region of interest is highlighted for the interest of the readers.

**Line 589-598:** We have provided more explanation of Case 2, the non-columnar virga event by adding the following paragraph.

The radiosonde vertical profiles reveal the absence of strong winds on the surface and within the boundary layer in the case of non-columnar virga. The 12 UTC profile overlaps with the virga's initial phase, whereas the 0 UTC profile provides information after the virga has ended (Figure 8b). Both soundings indicate the presence of dry layers between 900 and 600 mb and 600 and 300 mb, respectively. In contrast to the columnar virga, the temperature profile at 0 UTC follows the adiabatic ascent curve between 880 to 600 mb. Figure 8a shows that the surface wind and gust intensity were lower than what was observed in the columnar case. Based on ground measurements, we can see in Figure 8c that the surface was relatively more humid than in the columnar case and that the percent relative humidity increased rapidly at the end of the virga event. Based on the evidence presented above, we hypothesize that the virga intensity was lower in the non-columnar case, resulting in mild winds at the surface and as shown in Figure 8d, hence relatively lower concentrations of fine aerosol loading into the atmosphere. Furthermore, the higher moisture content in the air resulted in lower PM concentrations.

The paper is worthy of review, and upon making revisions, has potential for publication in AMT.

Thank you for your valuable comments and suggestions. It helped us enhance the content of the manuscript.

We would like to express our gratitude to the referee for taking the time to review our first response. Your comments and suggestions have been invaluable in helping us improve the quality and clarity of our work. The referee comments/suggestions are in black, and our responses are in red; the listed line numbers (bold font style) refer to the lines of the revised manuscript where the new corrections are observed.

This paper uses ceilometer data to identify potential cases of virga in El Paso, TX. It looks at the impact of virga on PM2.5 concentrations, where certain (columnar virga) cases at times led to surface enhancements of PM2.5 while others (non-columnar virga) did not.

The authors' response to initial reviewer feedback updated sections related to Case 1 (lines 265-290; columnar virga event with a dry microburst) and Case 2 (lines 320-345; non-columnar virga event) of the originally submitted manuscript. Both of those updates, including their updated figures, provide clarity, and improve the manuscript.

Thank you for taking the time to review our response to your first review report.

Given the relatively brief time scales involved in the outflow of microbursts, the statistical analysis of the influence on PM2.5 by potential virga-induced microbursts can be strengthened by using 5-minute averaged data from the continuous air monitoring stations network rather than hourly averaged data.

We agree with the referee, however, we tried hard and were unable to get access to the finer resolution dataset.

1. Does the paper address relevant scientific questions within the scope of AMT?
Yes

2. Does the paper present novel concepts, ideas, tools, or data?
Yes

3. Are substantial conclusions reached?
For the two highlighted case study examples, yes. For the analysis of the 50 cases over the 7 year period, using higher temporal resolution surface data would help substantiate possible correlations between wind gusts resulting from virga-induced dry microbursts and enhanced PM2.5 concentrations.

4. Are the scientific methods and assumptions valid and clearly outlined?
The methods are clearly outlined.

5. Are the results sufficient to support the interpretations and conclusions?

In general, yes. There is a note described below that

6. Is the description of experiments and calculations sufficiently complete and precise to allow their reproduction by fellow scientists (traceability of results)?
Yes, enough of a description is provided to allow other researchers to carry out similar studies.

7. Do the authors give proper credit to related work and clearly indicate their own new/original contribution?
Yes

8. Does the title clearly reflect the contents of the paper?
Yes

9. Does the abstract provide a concise and complete summary?
Yes

10. Is the overall presentation well structured and clear?
Yes, particularly with the updated portions provided in the authors' response to initial reviewer feedback.

11. Is the language fluent and precise?
With minor suggested edits (provided) and the improvements in the descriptions of updated sections in the authors' response to initial reviewer feedback, it is on the right track.

12. Are mathematical formulae, symbols, abbreviations, and units correctly defined and used?
In the updated versions of Figure 6b and Figure 8b provided in the authors' response to reviewers, please use m/s for the wind speed and max gust instead of miles/hr. Figures 6a, 6c and 8a, 8c, use Fahrenheit whereas the Skew-T plots (Figures 6b and 8b) use Celcius. Suggest using Celsius for all.

Figure 6a (**Line 355**), Figure 6c (**Line 365**), Figure 8a (**Line 565**) and Figure 8c (**Line 575**) are revised and appropriate units as suggested by the referee. The figure captions are also revised to reflect the above-mentioned changes.

13. Should any parts of the paper (text, formulae, figures, tables) be clarified, reduced, combined, or eliminated?
Aside from the minor note above regarding units, the updated versions of Figure 6 and Figure 8 provided by the authors' response to initial feedback address this.

14. Are the number and quality of references appropriate?
Yes

15. Is the amount and quality of supplementary material appropriate?

N/A

Comments and Suggestions:

1. Line 150: "a few miles away from the study site": clearly specify the distance in kilometers and the directional heading. CAMS 12 is later mentioned with a lat/lon provided, would make sense to do the same for the NWS Santa Teresa sounding site location.

**Line 172-173:**

The following revision is made to the manuscript:

Radiosonde observations were obtained from the nearest National Weather Service (NWS) (31°52'33" N, 106°36'39" W) located at Santa Teresa, New Mexico, 21 km away from the study site.

2. This paper provides a way of identifying virga events using ceilometer data. Confidence in proper identification of virga events will be higher for some cases than others. What suggestions do the authors have for other researchers to further confirm virga events? This is just a suggestion for consideration and the authors should not feel compelled to include this: The authors' could add suggestions for future researchers to help identify such cases and their influence on PM2.5 surface concentrations, such as (1) having an all-sky camera on the roof that saves one picture each minute, (2) releasing collocated radiosondes on demand during such events, and (3) obtaining higher temporal resolution data from surface monitors. Granted, this would require some additional funding but could provide things to consider for those who perform similar studies.

Thank you for the excellent suggestions. We do appreciate it. The last paragraph which describes the future scope of the work is revised to include the suggestions from the referee.

**Line 662-672:**

The availability of higher temporal resolution ground measurements will undoubtedly improve and solidify the correlation between the various parameters discussed in Table 1. If funds are available, using an all-sky camera (capable of capturing finer temporal resolution images) in conjunction with the ceilometer would greatly aid in capturing the virga precipitation. During the virga event, launching collocated radiosondes could provide an excellent dataset of vertical atmospheric profiles, especially the wind flows. A comprehensive study that includes such instrumentation and approaches would allow researchers to investigate the possibility of a strong connection between vertical winds, virga, and a local rise in PM levels. It also emphasizes the importance of having diverse instrumentation at the El Paso site, such as sonic anemometers, wind profilers, and barometers, which will provide a comprehensive dataset that will further enhance our understanding of virga and dry microbursts in the region. This research work will undoubtedly be a starting point for researchers to better comprehend the link between virga events and air quality. It will be worth analyzing the impact virga has on the climatology of precipitation in the semi-arid region.

3. Using 5-minute instead of hourly data for the analysis in Table 1 was suggested in the initial review response and the authors responded:

The 50 virga episodes that were classified in this work are unevenly distributed throughout the study period of 7 years. Since the 5-minute data was not available to us we decided to use the one-hour average data which is readily available in the public domain. We are grateful for your suggestion and will seek the 5-minute data for future use.

The authors' response is understandable, but it would be a missed opportunity to further improve this manuscript. The dry microburst events are short-lived and the 5-minute data may strengthen the analysis.

We agree with the referee, however, we tried hard and were unable to get access to the finer resolution dataset. Based on the availability, future work will include higher temporal resolution data.

Minor suggested edits that do not change authors' intended meaning:
Line 30: "before even reaching" → "before reaching"

**Line 30:** The sentence has been revised.

Line 35: "reviewed earth's climate system" → "reviewed Earth's climate system"

**Line 36:** The correction is made.

Lines 47-48: "radar and lidar, both ground-based, airborne, and satellite observations" please rewrite for clarity

**Line 47-49:** The following correction is made
Previous studies investigated it using remote sensing instruments such as ground-based or airborne radar and/or lidar, and in some cases satellite observations.

Line 48: "(Wang et al., 2018)" → "Wang et al. (2018)"

**Line 49:** The citation has been revised.

Lines 51-52: "(Saikranthi et al., 2014)" → "Saikranthi et al. (2014)"

**Line 53:** The citation has been revised.

Lines 53-54: "(Airey et al., 2021)" → "Airey et al. (2021)"

**Line 54:** The citation has been revised.

Line 70: "(Jullien et al., 2020)" → "Jullien et al. (2020)"

**Line 80:** The citation has been revised.

Link 71: "(Grazioli et al., 2017)" → "Grazioli et al. (2017)"

**Line 81:** The citation has been revised.

Line 74: Use the degree symbol (in Word, go to 'Insert' and then 'Symbol') rather than a superscripted "0"

**Line 84:** Symbol revised

Line 79: "(Tost et al., 2016)" → "Tost et al. (2016)"

**Line 89:** The citation has been revised.

Line 94: Change the semicolon to a period (or could use semicolons throughout that list of the different section topics).

**Line 103-106:** Revision made

Line 100: "doppler" → "Doppler"

**Line 118:** Revised

Line 102: Use the degree symbol rather than a superscripted "0"

**Line 120:** Symbol revised

Line 104: Delete "urban"

**Line 122:** Revised

Line 105: "Sun city's" → "Sun City's"

**Line 123:** Revised

Line 113: "aerosol layer height which can" → "aerosol layer height, which can"

**Line 131:** Addressed

Line 121: "CL31 was installed" → "A CL31 was installed"

**Line 139:** Addressed

Line 122: "CL51 has a" → "The CL51 has a"

**Line 140:** Addressed

Lines 135-136: "planetary boundary layer heights (Schafer et al. 2004) from the measured attenuated backscatter profiles." → "planetary boundary layer heights from the measured attenuated backscatter profiles (Schafer et al. 2004)."

**Line 159:** Addressed

Line 137: "for detection" → "for the detection"

**Line 160:** Addressed

Line 140: "doppler" → "Doppler"

**Line 163:** Addressed

Line 142: "diamter" → "diameter"

**Line 165:** Revised

Figure 2 caption: "We can observe lighter precipitation at the research site UTEP shown in red dot." → "We can observe light precipitation at the research site (UTEP) shown by the red dot."

**Line 235:** Addressed

Line 216: "years' " → "years" (no apostrophe)

**Line 253:** Addressed

Line 216: "at UTEP" → "at the UTEP"

**Line 253:** Revised

Line 218: "section 3" → "Section 3"

**Line 255:** Addressed

Line 227: "Similarly, to 2021, most" → "Similar to 2021, most"

**Line 264:** Addressed

Lines 228-229: "2015 saw the least number of virga events" → "2015 had the fewest virga events"

**Line 265:** Addressed

Figure 4 caption is missing a period at the end of it.

**Line 292:** Addressed

Line 256: "Case study 01" → "Case study 1"

**Line 301:** Addressed

Line 259: (till 22:45 UTC) → "(until 22:45 UTC)"

**Line 305:** Addressed

Line 263: "During the virga episode cloud base appears to be well above 4 km." → "During the virga episode, the cloud base appears to be well above 4 km."

**Line 308:** Addressed

Line 312: "ground around 21 CST" → "ground at approximately 21 CST" (and from the authors' response to reviewers the times throughout may end up in UTC)

**Line 450-463:** Addressed

Line 354: "(Table 1)" → "Table 1"

**Line 603:** Addressed

Line 354: "calculated p-value" → "calculated the p-value"

**Line 603:** Revised

Line 355: "P values" → "p-values"

**Line 604:** Addressed

Lines 355-356: "higher than 0.05. Which means" → "higher than 0.05; meaning"

**Line 604:** Addressed

Line 357: "vs" → "vs."

**Line 606:** Addressed

Lines 357-358: "had higher R-squared value in" → "had a higher R-squared value in"

**Line 606:** Addressed

Line 358: "indicate" → "indicates"

**Line 607:** Addressed

Lines 372-373: "We significantly extended our research and investigated the virga's impact on ground level PM concentrations." → "Our research extends the literature by providing an initial investigation into virga's impact on ground level PM concentrations." (or something to that effect)

**Line 634-635:** Revised

Line 373: "Cl31" → "CL31"

**Line 635:** Addressed

Line 389: "March 10, 2019," → "10 March 2019" (for consistency in how dates are formatted and is something to check throughout, most have DD Month YYYY)

**Line 654:** Modified and similar other date format typos rectified.

Line 403: "in the semi-arid region." → "in semi-arid regions."

**Line 672:** Addressed

Minor suggested edits the authors' response to initial reviewer feedback:
Figure 5 caption: "around 20-22 UTC" → "approximately 20-22 UTC" or "~20-22 UTC"

**Line 315:** Addressed

End of Page 5: "virga in case 1 was" → "virga in Case 1 was"

Addressed

Last page: "(figure 8b)" → "(Figure 8b)"; "figure 8 c" → "Figure 8c"; "figure 8d" → "Figure 8d"

**Line 589-598:** Addressed

Once again, we appreciate your time and expertise.